# *MECOM*-Rearranged Acute Myeloid Leukemia: Pathobiology and Management Strategies

**DOI:** 10.3390/hematolrep17060059

**Published:** 2025-10-31

**Authors:** Utsav Joshi, Rory M. Shallis

**Affiliations:** Department of Malignant Hematology, Moffitt Cancer Center, Tampa, FL 33612, USA; utsav.joshi@moffitt.org

**Keywords:** acute myeloid leukemia, AML, *EVI1*, inv(3), inversion 3, *MDS1*, *MECOM*, t(3;3)

## Abstract

Acute myeloid leukemia (AML) is an aggressive clonal hematopoietic malignancy, characterized by marked biological heterogeneity and variable clinical outcomes. Among its rarer genetic subsets is AML with rearrangements of the *MDS1* and *EVI1* complex locus (*MECOM*), occurring in fewer than 2% of newly diagnosed cases. This review examines the biology and clinical significance of *MECOM*-rearranged AML, with a focus on its diverse mechanisms of leukemogenesis, including chromosomal inversion and translocation involving 3q26. We discuss how aberrant *EVI1/MECOM* activity alters gene expression networks and drives malignant transformation. Current therapeutic approaches—including intensive chemotherapy, hypomethylating agents in combination with venetoclax, and allogeneic stem cell transplantation—are evaluated with particular emphasis on inv(3) and other t(3q26) subtypes. Despite these treatment strategies, outcomes remain poor, underscoring the urgent need for novel, more effective therapies for this high-risk form of AML.

## 1. Introduction

Acute myeloid leukemia (AML) is an aggressive clonal hematopoietic neoplasm characterized by maturing arrest with the consequent proliferation of leukemic blasts and the impairment of normal hematopoiesis [1]. Recent advances in genomics have established biological profiling as a cornerstone in risk stratification, informing the selection of patients for allogeneic hematopoietic cell transplantation (alloHCT), and directing contemporary management approaches in AML [2,3]. The rearrangement of the myelodysplasia syndrome 1 (*MDS1*) and ecotropic viral integration site 1 (*EVI1*) complex locus (*MECOM*) gene represents an uncommon AML-defining genetic abnormality that is present in less than 2% of all new cases [4,5]. *MECOM* is a zinc finger transcription factor normally expressed in hematopoietic stem cells (HSC) and essential for self-renewal. It influences several key regulators of hematopoiesis, including *GATA2*, *PBX1*, and *PML*, and thereby promotes malignant differentiation of myeloid precursors via its rearrangement with several partner genes [6]. In a large cohort of 3251 AML patients from the Swedish Adult Acute Leukemia Registry, 1% were found to carry inv(3)(q21q26) or t(3;3)(q21;q26), with the majority (52%) belonging to the 40–59-year age group [7]. Patients with *MECOM*-rearranged AML often show an aggressive disease course and poor survival, with a less-than-10% rate of 5-year overall survival (OS) [6,8,9]. Given its overall dismal prognosis, *MECOM*-rearranged AML is deemed an adverse risk disease subtype as per the European LeukemiaNet (ELN) 2022 risk stratification system [3]. This review aims to delineate the pathobiology of AML with *MECOM* rearrangements, evaluate current therapeutic strategies in the context of concomitant cytogenetic and molecular abnormalities, and present our treatment perspective while highlighting future directions for the management of this rare AML subtype.

## 2. Pathophysiology of *MECOM*-Rearranged AML

### 2.1. The MECOM Gene

The *MECOM* gene located at chromosome 3q26 was initially discovered as a *EVI1* isoform in murine leukemia model [10]. *EVI1*, the short isoform, is a zinc finger transcription factor that plays a key role in the development of HSC. *MDS1-EVI1*, the longer fusion isoform, is generated Via alternative splicing of the third exon of *MDS1* to the second of *EVI1* and has proline rich domain with methyltransferase activity that can alter transcriptional regulation and chromatin remodeling [11,12,13,14]. A third variant of *EVI1*, *EVI1∆324*, lacks part of zinc finger domain and, therefore, cannot bind DNA or promote AML leukemogenesis [14].

*MECOM* has been shown to function as both a transcriptional activator and repressor, influencing diverse targets such as Spi-1 Proto-Oncogene (*Spi1*), ETS Transcription Factor (*Erg*), Runt-Related Transcription Factor 1 (*RUNX1*), Phosphatase And Tensin Homolog (*PTEN*), and CCAAT Enhancer Binding Protein Alpha (*CEBPA*) [13,15,16,17,18]. It also modulates key signaling pathways, including repression of Transforming Growth Factor Beta (*TGF-β*) and Nuclear Factor Kappa B (*NFκB*) and activation of Notch Receptor (*NOTCH*) [13,19,20]. The multifaceted interaction of *EVI1/MECOM* with various gene targets promotes leukemogenesis (Figure 1). The upregulation of *MECOM* can occur due to inversion, translocation or, rarely, insertion involving chromosome 3q26, resulting in unchecked differentiation and the survival of myeloid precursors.

### 2.2. AML with inv(3)(q21.3q26.2) or t(3;3)(q21.3;q26.2)

One of the most common rearrangements involving *MECOM* is inv(3)(q21q26) or t(3;3)(q21;q26) [21]. Inv(3) is characterized by two breakpoints that occur on the same chromosome 3 between ribophorin 1 (*RPN1*) and *GATA2* at 3q21 and upstream of the first exon of *MECOM locus* at 3q26. The paracentric inversion juxtaposes the *GATA2* distal hematopoietic enhancer (*G2DHE*) to the *MECOM* gene that subsequently activates *EVI1* and causes haploinsufficiency of *GATA2* with reduced expression—also called “enhancer hijacking” [22,23].

In t(3;3), one breakpoint occurs between the *MDS1* promoter and the first *EVI1* exon, and the other at 3q21. This rearrangement relocates the *G2DHE* and *RPN1* regulatory elements from one chromosome to a site downstream of the *MECOM locus* on the homologous chromosome, leading to *MECOM* overexpression through a mechanism analogous to that seen in inv(3) [24].

### 2.3. AML with Other MECOM Rearrangements

Atypical or non-classic *MECOM* rearrangements occur in approximately 1% of AML or MDS and can result in comparable level of *EVI1* expression to inv(3) AML [24,25]. These include but may not be limited to t(3;21)(q26;q22), t(3;12)(q26;p13), t(2;3)(p21–22;q26), t(3;17)(q26;q22), t(3;7)(q26;q21), t(3;8)(q26;q24), t(3;10)(q26;q21), t(3;6)(q26;q25), and t(3;5)(q26;q22) and involve partner genes like *RUNX1*, Ets Variant 6 (*ETV6*), *THADA* armadillo repeat containing (*THADA*), cyclin dependent kinase 6 (*CDK6*), V-Myc avian myelocytomatosis viral oncogene homolog (*MYC*), and AT-rich interaction domain 1B (*ARID1B*) [5,14,25,26]. Several of these rearrangements result in overexpression of *EVI1/MECOM* by co-opting enhancers from translocation partners. However, certain translocations, such as t(3;12)(q26;p13) and t(3;21)(q26;q24), generate chimeric transcription factors. In the case of t(3;12), the *ETV6* promoter drives aberrant expression and activity of *EVI1*, while t(3;21) produces the *RUNX1–MDS1–EVI1* fusion protein, which disrupts the regulatory networks of both *RUNX1* and *EVI1* [27,28].

## 3. Concurrent Chromosomal Abnormalities and Mutations

A retrospective analysis of 6515 patients with newly diagnosed AML identified 146 cases with 3q26 translocations. Monosomy 7 was observed in 66% of patients with inv(3)/t(3;3) AML and in 31% of those with another t(3q26) AML. Del(7q) was present in 3–4% of cases across both groups, while del(5q) occurred in 6–12% [25]. The long arm of chromosome 7 harbors several critical genes like Enhancer of Zeste Homolog 2 (*EZH2*), Mixed-Lineage Leukemia 3 (*MLL3*), and the cytoplasmic regulators Sterile Alpha Motif Domain–Containing Protein (*SAMD9* and *SAMD9L*), and loss of one copy results in the loss of tumor-suppressor function contributing to leukemogenesis [29]. In addition, complex karyotypes were identified in 21% of cases with inv(3)/t(3;3) AML and 17% of those with t(3q26) AML [25]. *NRAS* mutations are observed in approximately 28% of cases with inv(3)/t(3;3) AML and in 25% of those with another t(3q26) AML [25]. Additional alterations involving the Rat Sarcoma viral oncogene homolog (*RAS*) signaling pathway, including mutations in Protein Tyrosine Phosphatase Non-Receptor Type 11 (*PTPN11*) and Neurofibromin 1 (*NF1*), have also been reported [30]. Furthermore, mutations in transcription factors such as *RUNX1* and IKAROS Family Zinc Finger 1 (*IKZF1*), as well as in epigenetic regulators including ASXL Transcriptional Regulator 1 (*ASXL1*), Splicing Factor 3b Subunit 1 (*SF3B1*), and U2 Small Nuclear RNA Auxiliary Factor 1 (*U2AF1*), are observed in more than 20% of patients with inv(3)/t(3;3) AML [24,30,31].

## 4. Diagnosis

### 4.1. Classification

In 2022, two new classification systems for the diagnosis and classification of AML were published: the 5th edition of the World Health Organization Classification (WHO5) and the International Consensus Classification (ICC) [2,32]. WHO5 classifies all *MECOM* rearrangement as AML-defining genetic abnormalities with no specific blast enumeration required to make a diagnosis. In contrast, although the ICC classifies inv(3)/t(3;3) and other *MECOM* rearrangements as AML-defining recurrent genetic abnormalities, this system requires >10% marrow or peripheral blood blasts to make a diagnosis.

The reduction or removal of blast percentage thresholds in recent AML classifications, including the ICC and WHO frameworks, reflects the recognition that *MECOM*-rearranged myeloid neoplasms exhibit aggressive biology and a high propensity for leukemic transformation, even at low blast counts [2,32,33]. Although earlier studies demonstrated no significant survival difference between patients with <20% (previously classified as MDS) and ≥20% blasts—supporting the removal of the arbitrary blast cut-off—a recent study suggests that blast percentage retains prognostic significance within *MECOM*-rearranged AML [33,34,35]. In this analysis, patients with <20% blasts had a composite complete remission (CRc) rate of 57% versus 45% for those with ≥20% blasts (*p* = 0.31), and a median overall survival (OS) of 17 months compared with 9 months (*p* < 0.01). Notably, cases with <5% blasts at presentation showed a median OS of 63 months. When stratified by <10%, 10–20%, or ≥20% blasts, outcomes were similar between the <10% and 10–20% groups (*p* = 0.84), but patients with ≥20% blasts had significantly inferior OS compared with both <10% (*p* = 0.0017) and 10–20% (*p* = 0.0022) cohorts [35]. These findings indicate that, while the blast percentage threshold may no longer serve a diagnostic purpose, higher blast counts within *MECOM*-rearranged AML are associated with more aggressive disease biology and poorer survival outcomes.

### 4.2. Clinical Features

Like other subtypes of AML, cases with *MECOM* rearrangements typically present with anemia, thrombocytopenia, and either leukopenia or leukocytosis. Subtle clinical differences, however, have been noted when comparing these patients to those with non-3q26 AML, as well as among different *MECOM* cytogenetic subgroups. No significant differences in sex distribution are observed across inv(3)/t(3;3), other *MECOM*-rearranged AML, and non-3q AML. Likewise, age at diagnosis is comparable between inv(3)/t(3;3) and other *MECOM*-rearranged cases, although both groups tend to present at a younger age than patients with non-3q AML [25,36]. Patients with inv(3)/t(3;3) AML more often exhibit higher white blood cell and platelet counts relative to non-3q AML [25]. In contrast, those with pericentric inv(3) are more likely to present with monocytosis in both peripheral blood and bone marrow, accompanied by thrombocytopenia and reduced megakaryocytes, when compared with patients with the classic paracentric inv(3)/t(3;3) [37]. Morphologically, dysplasia is most frequently observed in the megakaryocytic lineage, followed by the erythroid lineage [36].

### 4.3. Diagnosis

Accurate diagnosis of *MECOM*-rearranged AML relies on the detection of the lesion and thus a combination of cytogenetic and molecular methods. Conventional G-banding analysis usually represents the first step in detecting structural cytogenetic patterns like inv(3) or t(3;3). However, this method is limited to detecting aberrations at least 1 Mb in size and cryptic structural variants may be missed [38]. In one study, pericentric inv(3) was missed in 16 of 17 cases by conventional karyotyping and was subsequently detected using metaphase Fluorescence In Situ Hybridization (FISH) [37]. FISH testing, particularly with a *MECOM* break-apart probe (irrespective of the partner gene involved), is the most used adjunct to routine karyotype analysis [39]. It enables detection of cryptic or atypical variants, complex chromosomal rearrangements that appear as unidentifiable marker chromosomes, and *MECOM* amplifications such as segmental duplications [6,40,41]. However, FISH is limited to predefined loci and may not identify novel fusion partners. Polymerase chain reaction (PCR)–based assays, including allele-specific PCR, real-time PCR (qPCR), and digital PCR (dPCR), provide greater sensitivity, with dPCR capable of detecting variants at levels as low as 10^−4^ [42]. Advances in next-generation sequencing (NGS) technologies now offer the most comprehensive approach, allowing identification of both canonical and novel *MECOM* fusion genes, as well as cryptic 3q26 rearrangements. With error-correction methods such as unique molecular identifiers (UMIs), NGS sensitivity can approach 10^−6^, minimizing false positives in low variant allele frequency (VAF) settings [42]. NGS remains costly, time-consuming, and dependent on sophisticated bioinformatics pipelines, but improvements in these domains over time may yield more prevalent use.

In practice, a step-by-step diagnostic strategy is recommended, particularly in resource-limited settings where it may not be feasible to utilize all the above diagnostic modalities. This approach involves initial screening with G-banding analysis, followed by reflex *MECOM* FISH for suspected or inconclusive cases, and confirmation with qPCR or RNA-based NGS to define fusion partners and concurrent mutations for accurate risk stratification and therapeutic planning.

## 5. Response Rates in inv(3)/t(3;3) AML

Standard frontline therapies for AML include intensive chemotherapy regimens incorporating cytarabine with anthracyclines, cladribine, or fludarabine, as well as less-intensive approaches using hypomethylating agents (HMA) in combination with venetoclax. However, *MECOM*-rearranged AML has consistently demonstrated poor responsiveness to these treatment strategies (Table 1). In one of the largest cohorts of ninety-four patients, the complete remission (CR) rate following intensive induction was 31% [25]. Across other series, CRc rates with intensive chemotherapy ranged from 15% to 36%, reinforcing the observation that fewer than half of patients with this cytogenetic subgroup achieve remission with standard chemotherapy [43,44,45,46,47]. In a separate study evaluating both newly diagnosed and relapsed/refractory (R/R) cases, the CRc rates were 46% in newly diagnosed and 20% in R/R cases, with 30-day mortality of 14% and 5%, respectively. When compared with outcomes from less-intensive regimens, the CRc rates were not significantly different, at 47% and 8% for patients with newly diagnosed and R/R disease, respectively (both *p* > 0.05) [48].

Given the higher incidence of adverse events with intensive chemotherapy and the lack of significant differences in response rates, less-intensive regimens such as HMA, with or without venetoclax, represent a reasonable therapeutic alternative for inv(3)/t(3;3) AML. Reported CR/CRc rates with HMA-based therapies range from 24% to 36% [48,49,50]. In a small series by Merz Et Al., the CRi rate reached 50%, although the limited sample size warrants caution in interpretation [45]. HMAs appear to be the primary driver of response, with CRc rates of 36% in patients treated with HMA compared with 10% in those who did not receive HMA. This observation may also have a biologic explanation, as *MECOM* overexpression is associated with aberrant hypermethylation signature mediated through interactions with DNA methyltransferases (*DNMT*), providing a rationale for the use of HMA (*DNMT* inhibitors) in AML with inv(3)/t(3;3) [51]. The addition of venetoclax provided minimal incremental benefit, with CRc rates of 33% versus 31% for regimens with and without venetoclax, respectively. Moreover, remissions with venetoclax-containing therapy were short-lived, with a median CR duration of 5.4 months and no sustained responses beyond 6 months [48]. Collectively, these findings suggest that HMA monotherapy may be an appropriate treatment option in this high-risk AML subset. 

**Table 1 hematolrep-17-00059-t001:** Response rate and survival outcomes in inv(3)/t(3;3).

Study	N	Inv(3)/t(3;3), N(%)	Treatment Modalities	Response for inv(3)/t(3;3)	PFS/RFS/EFS/DFS for inv(3)/t(3;3)	OS: for inv(3)/t(3;3)
Annunziata et al. (2012) [52]	16	6 (38)	IC		median DFS: 4 months	median OS: 6 months
Boussi et al. (2025) [49]	43	22 (52)	HMA, Venetoclax	CRc 33% *		
				ND 36%		ND-median OS: 7.2 months *
				R/R 29%		R/R-median OS: 8.4 months *
Grimwade et al. (2010) [43]	5876	69 (1)	IC	CR 36%		10-year OS: 3%
Halaburda et al. (2018) [53]	98	98 (100)			2-year leukemia free survival 20%	2-year OS: 26%
Kusne et al. (2023) [46]	60 ^#^	39 (65)	IC, HMA, Venetoclax	CR 40% *; 36% IC, 4% LI		median OS: 11 months *
Lugthart et al. (2010) [25]	6515	94 (32)	IC (7 + 3 +/− etoposide)	CR 31%	5-year RFS: 4.3%, 5-year EFS: 0%	median OS: 10.3 months, 5-year OS: 5.7%
Merz Et Al. (2020) [45]	15	15 (100)	HMA, lenalidomide, IC (7 + 3, HiDAC, FLAG-Ida)	ORR 33.3%		
				HMA/lenalidomide—ORR 75%, CRi 50%, MLFS 25%	HMA/lenalidomide—median EFS: 5.2 months	HMA/lenalidomide—median OS: 9.3 months
				Other IC—ORR 21.4%, CR 7.1%, CRi 7.1%, MLFS 7.1%	Other IC—median EFS: 0.72 months	Other IC-median OS: 8.3 months
Moscvin et al. (2023) [47]	40	40 (100)	IC, LI	CR 22%		
Ouyang et al. (2025) [54]	90	90 (100)				median OS: 8 months, 5-year OS: 15.29%
Polprasert et al. (2023) [8]	9 ^#^	5 (55)	IC, HMA	CR 0% * (AML only)		1-year OS: 16.7% * (AML only)
Richard-Carpentier et al. (2023) [48]	108 (96 evaluable)	96 (100)	IC, HMA, Venetoclax	CRc 31%		
	ND-53		ND: 67% IC, 33% LI	ND-46% IC, 47% LI	ND-median RFS: 3.3 months	ND-median OS: 7.9 months, 3-year OS: 8.8%
	R/R-55		R/R: 45% IC, 55% LI	R/R—20% IC, 8% LI	R/R-median RFS: 2.1 months	R/R-median OS: 5.9 months, 3-year OS: 7.1%
Sallman et al. (2020) [50]	411 ^#^	49 (12)	IC, HMA	CR 24%		median OS: 12.7 months
Sitges et al. (2020) [44]	61	61 (100)	IC	CR 29%		median OS: 8.4 months, 1-year OS: 41%, 4-year OS: 13%
Tang et al. (2023) [37]	17 ^Ω^	17 (100)	Chemotherapy, GO, FLT3 inhibitors, HCT, clinical trials	Partial remission 7%		median OS: 4–5 months from the date of inv(3) detection
Wanquet et al. (2015) [9]	157 ^#^	40 (25)	Azacitidine			median OS: 10 months, 1-year OS: 46.4%
Jen et al. (2025) [35]	152	87 (57)	IC, HMA, Low dose cytarabine, Venetoclax	ORR 50%, CR 26%		median OS: 9 months, 3-year OS: 11%

^#^ Includes both AML and MDS, ^Ω^ Includes 1 case each of MDS and CMML, * Includes both inv(3)/t(3;3) and other 3q26, IC: intensive chemotherapy, HMA: hypomethylating agents, LI: low intensity, PFS: progression free survival, RFS: relapse free survival, EFS: event free survival, DFS: disease free survival, OS: overall survival, ORR: overall response rate, CR: complete response, CRc: composite clinical response, CRi: complete response with incomplete count recovery, MLFS: morphologic leukemia free state, ND: newly diagnosed, R/R: relapsed/refractory, HiDAC: high dose cytarabine, FLAG-IDA: Fludarabine, Cytarabine, Idarubicin and G-CSF, FLT3: fms-related tyrosine kinase 3, GO: Gemtuzumab ozogamicin, HCT: hematopoietic cell transplant.

## 6. Overall Survival in inv(3)/t(3;3) AML

The relatively short duration of remission drives the poor survival outcomes for inv(3)/t(3;3) AML across treatment modalities (Table 1). In the Lugthart cohort, median OS was 10.3 months, with a 5-year OS of only 5.7% [25]. Similarly, Grimwade Et Al. reported 10-year OS of 3%, highlighting the near absence of long-term survivors [43]. Among patients treated with intensive chemotherapy, median OS ranged from 4 to 10 months, while studies of HMA-based therapy reported median OS between 7 and 12 months [9,25,37,44,45,49,50,52,54]. Although response rates vary, survival outcomes appear comparable in newly diagnosed and R/R settings, with newly diagnosed patients demonstrating a relapse-free survival (RFS) of 3.3 months and median OS of 7–8 months, compared with an RFS of 2.1 months and median OS of 6–8 months for R/R disease [48,49]. Collectively, these studies underscore that inv(3)/t(3;3) AML is characterized by both low remission rates and survival that is amongst the shortest in cytogenetically defined AML.

## 7. Response Rates and Overall Survival in Other 3q26 AML

In contrast to inv(3)/t(3;3) AML, patients with disease harboring other 3q26 rearrangements can expect higher rates of initial response to therapy, although outcomes remain suboptimal overall (Table 2). In the Lugthart Et Al. series (N = 52), the CR rate was 44%, compared with 31% in inv(3)/t(3;3) [25]. Grimwade Et Al. reported an even higher CR rate of 59% in a cohort of 108 patients [43]. Notably, data for HMA in this subgroup are limited, with one retrospective study of patients with MDS (with majority having increased blasts and treated in the pre-venetoclax era), reporting a remission rate of 25% [50].

OS in patients with AML harboring other 3q26 abnormalities, while marginally superior to inv(3)/t(3;3), remains poor compared with non-3q AML (Table 2). In a large cohort treated with intensive chemotherapy, Lugthart Et Al. reported a median OS of 13.7 months and a 5-year OS of 29.7% [25]. Grimwade Et Al. reported a 10-year OS of 11.3%, again highlighting the rarity of durable remission despite higher induction response rates [43]. Annunziata Et Al. observed a notably worse median OS of 3 months; however, interpretation is limited by the small sample size (N = 3) [52]. For patients receiving HMA (including that for disease with <10% blasts), median OS ranged from 10 to 18 months, further illustrating the challenges in achieving sustained remission in this group [9,50]. Collectively, published survival analyses consistently demonstrate that patients with non-inv(3)/t(3;3) 3q26 AML, although experiencing slightly improved initial responses compared to the inv(3)/t(3;3) subgroup, continue to face poor long-term outcomes. These data reinforce the adverse prognostic significance of *MECOM* rearrangements in AML, irrespective of the associated partner chromosome. 

## 8. Prognosis

Prognostic outcomes in *MECOM*-rearranged AML are influenced by a range of clinical, cytogenetic, molecular, and treatment factors [9,46,48]. Importantly, achieving CR at HCT has consistently emerged as a favorable prognostic factor [48,53]. The presence of additional cytogenetic abnormalities predicts worse survival compared with cases in which *MECOM* rearrangement is the sole abnormality [55]. The inferior outcome is particularly pronounced with the presence of monosomy 7 or 7q deletion [25,44,47,55]. *MECOM* rearrangement co-existing with a complex karyotype is associated with significantly worse OS when compared with *MECOM* rearrangement alone or those with non-complex karyotypes [35]. Additional adverse prognostic indicators include elevated WBC (≥20 × 10^9^/L) and secondary or therapy related AML [48]. Molecularly, an increased mutational burden is unfavorable, as patients with ≥2 mutations had significantly shorter survival compared with those harboring ≤1 mutation [55]. Specific high-risk mutations such as *ASXL1*, *NRAS* and *TP53* mutations showed independent association with inferior OS across multiple studies [48,55]. AlloHCT may provide a survival advantage similar to other ELN adverse-risk AML and is discussed in more detail below.

In a study involving 152 patients with *MECOM*-rearranged AML, baseline bone marrow blast percentage was found to be significantly associated with overall survival. The most favorable outcomes were observed in patients with non-inv(3)/t(3;3) 3q26 AML and blast counts below 20%. Overall survival was significantly better in the 3q26 AML group with <20% blasts compared to those with ≥20% blasts (*p* = 0.0007) and to patients with inv(3)/t(3;3) AML and ≥20% blasts (*p* = 0.0001), but was comparable to that of inv(3)/t(3;3) AML with <20% blasts (*p* = 0.30). Intensive chemotherapy was associated with a higher hazard of death compared to low-intensity therapy, although the lower bound of the confidence interval approached unity. The addition of venetoclax did not significantly affect OS [35].

## 9. Allogeneic HCT

AlloHCT at first complete remission (CR1) is a recommended consolidative strategy for adverse-risk AML, including those cases characterized by *MECOM* rearrangements. Despite this strategy, response rates—encompassing CR and CRc—range between 20 and 45% across published cohorts [9,46,47,48,56]. Details of the studies are presented in Table 3. Importantly, alloHCT in CR1 is consistently associated with improved survival outcomes. One study demonstrated a 5-year OS of 44% in the alloHCT group compared to 6% among non-alloHCT patients. Relapse rates are reduced following transplant (2-year cumulative incidence of relapse [CIR] 57% with alloHCT versus 86% without), and RFS is similarly improved [48]. Interestingly, a study of 29 patients who underwent alloHCT evaluated the impact of transplant on survival by *MECOM* rearrangement type and blast percentage. Patients with <20% blasts at baseline had the most favorable outcomes, with median OS not reached and a 3-year OS of 66%. Among *MECOM* subtypes, those with non-inv(3)/t(3;3) 3q26-rearranged AML achieved the best post-alloHCT outcomes, with median OS not reached and 3-year OS 71% compared with a median OS of 14 months and 3-year OS of only 11% in patients who did not undergo alloHCT [35].

Nevertheless, long-term outcomes remain suboptimal, even in the context of early alloHCT, with persistent risk of relapse and disease-related mortality. Notably, substantial non-relapse mortality has been observed in some studies, underscoring the importance of achieving CR prior to alloHCT, as pre-transplant remission status was associated with reduced transplant-related risk [53]. Overall, these findings highlight that alloHCT in CR1 offers the greatest chance for durable survival in *MECOM*-rearranged AML, although further innovation is needed to meaningfully improve outcomes in this high-risk subgroup [46,47,48,53].

## 10. Emerging Therapeutic Options

Several novel therapeutic strategies are being explored in *MECOM*-rearranged AML, reflecting a rapidly expanding clinical landscape. One promising therapeutic target under study involves Bromodomain Containing 4 (*BRD4*), whose upregulation at the rearranged *GATA2* enhancer drives *EVI1* overexpression [57]. Targeting *BRD4* with a bifunctional small molecule, NICE-01 (AP1867-PEG2-JQ1), promotes differentiation of AML progenitor cells and subsequent cell death by recruiting transcriptional coactivators to *MECOM* and inducing targeted protein degradation [58,59]. Similarly, treatment with a bromodomain and extra-terminal motif (*BET*) protein inhibitor (BETi) suppresses *EVI1* expression and reduces the viability of *EVI1* overexpressing AML cells [60]. Another compound, tegavivint, disrupts the interaction of *EVI1* and *LEF1* with the scaffold protein *TBL1*, resulting in depletion of *EVI1* and loss of enhancer activity [60,61]. Importantly, preclinical studies have shown that combination of BETi and tigavivint produce synergistic lethality in AML models and represent another strategy to target AML [60].

Beyond *BRD4*, other molecular targets have been identified, including *KDM1A*, *p300*, *HDAC3*, *mTOR*, *PIK3CA*, and *XIAP* [62]. Preclinical studies have demonstrated that mivebresib (BETi), dactolisib (*mTOR/PI3K* inhibitor), and LCL161 (*XIAP* inhibitor) each induce dose-dependent apoptosis in *MECOM*-rearranged AML cell lines. Notably, combinations of these agents reduced leukemia burden and improved survival in experimental mouse models more effectively than single-agent therapy (*p* < 0.05) [62].

Another novel agent in development is AB8939, a tubulin inhibitor designed to circumvent two major mechanisms of resistance in AML—P-glycoprotein and myeloperoxidase [63]. In highly resistant cytarabine treated patient-derived xenograft (PDX) models, AB8939 alone or in combination with cytarabine significantly improved survival and reduced leukemic burden [63,64,65]. The agent is currently being tested in an ongoing phase 1/2 clinical trial (NCT05211570) for relapsed/refractory AML and MDS, where preliminary data show that two of the first four patients achieved responses in the single agent setting [66].

Poly(ADP-ribose) polymerase 1 (*PARP1*) has also been identified as a potential new target in *MECOM*-rearranged AML. *PARP1* has been implicated in oncogenic super-enhancer formation, with *PARP* inhibition leading to reduced *EVI1* expression and disruption of *EVI1–G2DHE* interactions [67]. Given the availability of FDA-approved *PARP* inhibitors, including *PARP1*-selective inhibitors that may mitigate myelosuppression, this represents a potentially translatable therapeutic approach pending validation in clinical trials [68].

Similarly, chromatin remodeling enzymes *BRG1* (*SMARCA4*) and *BRM* (*SMARCA2*), the core ATPases of the *BAF* complex, have been identified as druggable targets. The selective *BRG1/BRM* inhibitor FHD-286 has been shown to reduce *EVI1* protein levels, and when combined with decitabine, BETi (OTX015), or *HAT* inhibitor (GNE-781), exerted synergistic antileukemic effects in *MECOM*-rearranged AML cells [69]. All these studies highlight a dynamic and evolving therapeutic landscape, with multiple new druggable pathways under active investigation to address the profound unmet need in *MECOM*-rearranged AML.

## 11. Our Approach

*MECOM*-rearranged AML represents an informative example of why the AML biological evaluation, inclusive of sensitive cytogenetic techniques, is mandatory. The relatively short durations of response amongst patients fortunate enough to achieve one informs a critical need for expedient engagement with bone marrow transplant/cellular therapy colleagues, HLA typing and donor search—even before confirmation of response/remission—given the favorable, albeit insufficient data for early alloHCT while in CR1.

The optimal frontline induction strategy for *MECOM*-rearranged AML is unknown and debated. No more than half of intensively treated patients will enter CR after intensive induction. Less-intensive therapy with an HMA backbone is associated with a similar rate of response, with less treatment-related mortality, although with what appears to be lesser quality and depth of remission (i.e., less CR and more MLFS). The addition of venetoclax to HMA yields underwhelming improvement in response in retrospective studies and has been attributed to *MECOM*-rearranged AML’s overexpression of anti-apoptotic *BH3* family proteins. Although MLFS as best response allows patients to proceed to alloHCT, “better” quality responses characterized by varying degrees of count recovery have classically associated with better post-alloHCT outcomes. The role and prognostic impact of the pre-alloHCT MRD assessment is uncertain and requires further study. Owing to these gaps in knowledge and lack of data guiding a deviation from historical treatment paradigms, we still treat the intensive therapy-appropriate and alloHCT-intended patient with *MECOM*-rearranged AML with frontline intensive therapy. Subgroup analyses from ongoing, prospective, randomized trials comparing Intensive Vs. Less-Intensive approaches for higher-risk AML (NCT05554406, NCT04801797) may inform this preference further.

Preference for enrollment onto a clinical trial is the standard for all patients with adverse risk AML. Novel approaches have largely focused on the treatment of R/R *MECOM*-rearranged AML, and should an investigational product demonstrate promise in this setting, we expect a frontline combination trial targeting this population in need.

## 12. Conclusions

*MECOM*-rearranged AML represents a rare, but clinically distinct entity defined by characteristic biology, poor response to standard therapies, and dismal long-term survival. Despite the use of these therapies, including intensive chemotherapy and alloHCT, survival remains amongst the shortest of any cytogenetically defined AML subgroup. Encouragingly, advances in molecular profiling have identified novel targets, including *BET* proteins, *EVI1*-associated co-factors, epigenetic regulators, signaling pathways, and chromatin remodeling complexes. Early preclinical studies have also shown synergistic benefits with rational drug combinations. Translation of these findings into well-designed clinical trials will be essential to improve outcomes.

## Figures and Tables

**Figure 1 hematolrep-17-00059-f001:**
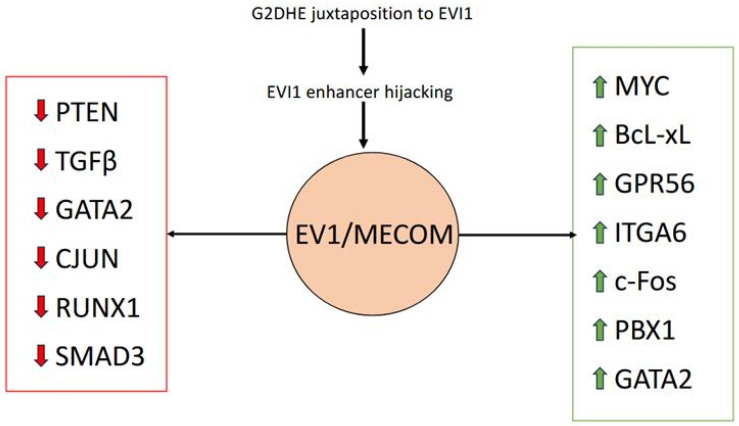
Effect of *EVI1/MECOM* upregulation on multiple transcriptional pathways. Red arrows denote downregulation and green arrows denote upregulation.

**Table 2 hematolrep-17-00059-t002:** Response rate and survival outcomes in AML with another t(3q26).

Study	Total N	Other 3q26, N (%)	Treatment Modalities	Response for Other 3q26	PFS/RFS/EFS/DFS for Other 3q26	OS for Other 3q26
Annunziata Et Al. (2012) [52]	16	3 (19)	IC			median OS: 3 months
Boussi et al. (2025) [49]	43	17 (40) ^%^	HMA, Venetoclax	CRc 33% *		ND-median OS: 7.2 months * R/R-median OS: 8.4 months *
Grimwade Et Al. (2010) [43]	5876	108 (2) ^%^	IC	CR 59%		10-year OS: 11.3%
Kusne et al. (2023) [46]	60 ^#^	7 (12)	IC, HMA, Venetoclax	CR 40% *		median OS: 11 months *
Lugthart Et Al. (2010) [25]	6515	52 (18)	IC (7 + 3 +/− etoposide)	CR 44%	5-year RFS: 45.9%, 5-year EFS: 15.8%	median OS: 13.7 months, 5-year OS: 29.7%
Polprasert et al. (2023) [8]	9 ^#^	4 (45)	IC, HMA	CR 0% * (AML only)		1-year OS: 16.7% * (AML only)
Sallman et al. (2020) [50]	411 ^#^	20 (5)	IC, HMA	CR 25%		median OS: 18 months
Wanquet et al. (2015) [9]	157 ^#^	36 (23)	Azacitidine			median OS: 10.1 months, 1-year OS: 48.7%
Jen et al. (2025) [35]	152	65 (43)	IC, HMA, Low dose cytarabine, Venetoclax	ORR 58%, CR 34%		median OS: 11 months, 3-year OS: 22%

^#^ Includes both AML and MDS, ^%^ Includes other 3q abnormalities, * Includes both inv(3)/t(3;3) and other 3q26, IC: intensive chemotherapy, HMA: hypomethylating agents, PFS: progression free survival, RFS: relapse free survival, EFS: event free survival, DFS: disease free survival, OS: overall survival, ORR: overall response rate, CR: complete response, CRc: composite clinical response, ND: newly diagnosed, R/R: relapsed/refractory.

**Table 3 hematolrep-17-00059-t003:** Response rate and survival outcomes in *MECOM*-rearranged AML undergoing HCT.

Study	OS	Response	RFS/DFS
Halaburda et al. (2018) [53]	2-year OS: 26% (95% CI, 17–35%)		2-year leukemia-free survival: 20% (95% CI, 11–24%)
Kusne et al. (2023) [46]	Median OS: 19 months	CR 36% (IC), 4% (non-intensive)	
Richard-Carpentier et al. (2023) [48]	5-year OS: 44% (95% CI, 20–96)	ORR: 40%	
Sitges et al. (2020) [44]	1-year OS 50% (95% CI, 34–66), 4-year OS 24% (CI 95%, 9–39)	CR: 75%	
Wanquet et al. (2015) [9]	Median OS: 21 months (95% CI, 6.1–35.9); 5-year OS: 10.5%		
Weisser et al. (2007) [56]	Median OS: 899 days, 2-year OS: 62%	ORR: 75%	
Jen et al. (2025) [35]	Median OS: 52 months (95% CI, 15-NR)		

IC: intensive chemotherapy, OS: overall survival, CI: confidence interval, CR: complete response, ORR: overall response rate, NR: not reached, HCT: hematopoietic cell transplant.

## Data Availability

All data supporting reported results can be found within the study.

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
