# Peer review of "MECOM-Rearranged Acute Myeloid Leukemia: Pathobiology and Management Strategies"

_hematolrep, 2025, doi:10.3390/hematolrep17060059_

Round 1
Reviewer 1 Report
I added two references, last with report of 125 cases from a large centre which not reported. I woulds like to see some Kaplan-Meier curves and more about transplantation. Another reference show true incidence in the population based registry.
See above.
Author Response
Comments:I added two references, last with report of 125 cases from a large center which not reported. I would like to see some Kaplan-Meier curves and more about transplantation. Another reference shows true incidence in the population-based registry.
Response: We thank the Reviewer for their valuable feedback. We have incorporated both recommended references into the manuscript and expanded the discussion accordingly. The following statements are now included:
- “MECOM rearrangement co-existing with a complex karyotype is associated with significantly worse OS when compared with MECOM rearrangement alone or those with non-complex karyotypes.”
- “In a study involving 152 patients with MECOM-rearranged AML, baseline bone marrow blast percentage was found to be significantly associated with overall survival. The most favorable outcomes were observed in patients with non-inv(3)/t(3;3) 3q26 AML and blast counts below 20%. Overall survival was significantly better in the 3q26 AML group with <20% blasts compared to those with ≥20% blasts (p=0.0007) and to patients with inv(3)/t(3;3) AML and ≥20% blasts (p=0.0001), but was comparable to that of inv(3)/t(3;3) AML with <20% blasts (p=0.30). Intensive chemotherapy was associated with a higher hazard of death compared to low-intensity therapy, although the lower bound of the confidence interval approached unity. The addition of venetoclax did not significantly affect OS.”
- “In a large cohort of 3,251 AML patients from the Swedish Adult Acute Leukemia Registry, 1% were found to carry inv(3)(q21q26) or t(3;3)(q21;q26), with the majority (52%) belonging to the 40–59-year age group.”
We have also updated our Tables section to reflect the above additions.
Regarding transplantation, we have updated the section to include the most current data on HCT use in MECOM-rearranged AML. In particular, we have added findings from Jen et al. (reference 1 suggested by the Reviewer) highlighting new insights into the impact of bone marrow blast percentage and MECOM rearrangement subtype on outcomes in transplanted patients—information not available in previously cited studies.
“Interestingly, a study of 29 patients who underwent alloHCT evaluated the impact of transplant on survival by MECOM rearrangement type and blast percentage. Patients with <20% blasts at baseline had the most favorable outcomes, with median OS not reached and a 3-year OS of 66%. Among MECOM subtypes, those with non-inv(3)/t(3;3) 3q26-rearranged AML achieved the best post-alloHCT outcomes, with median OS not reached and 3-year OS 71% compared with a median OS of 14 months and 3-year OS of only 11% in patients who did not undergo alloHCT.”
With respect to KM curves, we appreciate the importance of visual representation of survival data; however, including KM curves from multiple studies in a review article is practically challenging. We have summarized the key survival data narratively and in tables to maintain accuracy and cohesion within the manuscript.
Reviewer 2 Report
The presented article titled "MECOM-rearranged acute myeloid leukemia: pathobiology and management strategies" is very well written. It has a clear structure typical of review articles, with distinct sections such as introduction, pathophysiology, diagnostics, treatment, and conclusions. The text is based on the latest research and appropriately cited, which confirms its credibility. The article thoroughly discusses the biological complexity, clinical features, and current treatment options for this rare and difficult-to-treat AML subtype, as well as highlights knowledge gaps and the need for further research. The figures are of good quality and help readers understand the topic effectively.
However, including a broader clinical context or practical implications for the daily work of hematologists would enhance the article by better linking the scientific data to clinical practice. This would assist clinicians in translating complex molecular and biological insights into patient management strategies, ultimately improving care outcomes. I have a few minor comments.
- The article lacks the expansion of some abbreviations, such as NOTCH.
- The tables in the article do not explain all the abbreviations used within them.
- Please add references to the literature for each figure and table.
Author Response
Comments 1: The presented article titled "MECOM-rearranged acute myeloid leukemia: pathobiology and management strategies" is very well written. It has a clear structure typical of review articles, with distinct sections such as introduction, pathophysiology, diagnostics, treatment, and conclusions. The text is based on the latest research and appropriately cited, which confirms its credibility. The article thoroughly discusses the biological complexity, clinical features, and current treatment options for this rare and difficult-to-treat AML subtype, as well as highlights knowledge gaps and the need for further research. The figures are of good quality and help readers understand the topic effectively.
However, including a broader clinical context or practical implications for the daily work of hematologists would enhance the article by better linking the scientific data to clinical practice. This would assist clinicians in translating complex molecular and biological insights into patient management strategies, ultimately improving care outcomes. I have a few minor comments.
Response 1: We thank the reviewer for this constructive feedback. We have expanded the manuscript to include additional subsections discussing blast thresholds and their prognostic implications, a stepwise diagnostic strategy, and our practical approach to managing MECOM-rearranged AML. These additions aim to provide a clearer clinical context and bridge the gap between molecular insights and real-world practice and have further been discussed below in subsequent comments.
Comments 2: The article lacks the expansion of some abbreviations, such as NOTCH.
Response 2: Abbreviations have been expanded within the manuscript.
Comments 3: The tables in the article do not explain all the abbreviations used within them.
Response 3: Full forms of the abbreviations have been added in the tables.
Comments 4: Please add references to the literature for each figure and table.
Response 4: References have now been added in the tables and figures.
Reviewer 3 Report
This manuscript is an excellent and comprehensive review focusing on MECOM-rearranged AML, a highly aggressive hematologic malignancy. It covers diverse topics, ranging from basic pathophysiology and a review of conventional treatment outcomes to the introduction of novel small-molecule compounds that promise future therapeutic improvement. The work is highly commendable for citing numerous up-to-date references, making it an extremely useful resource for researchers and clinicians in this field.
While the current content is strong, we believe addressing the following four points will further enhance the academic rigor and clinical utility of the paper, positioning it as a definitive resource on this rare yet important disease. We look forward to receiving a revised manuscript.
Specific Comments and Suggestions for Revision
1.Strengthening the Discussion on Diagnostic Methods and the Risk of Misdiagnosis
Given the severe prognosis of MECOM-rearranged AML, accurate and timely diagnosis is paramount. We kindly suggest adding a more detailed discussion on the utility, limitations, and detection sensitivity of various diagnostic methods, including Conventional G-banding, FISH, RT-PCR, and Next-Generation Sequencing (NGS).
Specifically, please discuss the risk of missing cryptic fusions and propose an integrated diagnostic strategy or algorithm that combines these methods to ensure comprehensive detection in clinical practice.
2.Clarifying the Clinical Significance of the Blast Percentage Discrepancy (WHO/ICC)
The manuscript touches upon the AML classification criteria regarding the blast percentage threshold (10% in the selected text). We encourage you to provide a more in-depth discussion on the clinical implications and prognosis of $MECOM$-rearranged cases with blast counts below 10% (e.g., MDS/AML-MDS overlap).
Based on existing literature, please articulate the rationale for whether these cases behave differently from typical AML (blast $\ge 10\%$) or if the underlying poor pathobiology warrants uniform treatment with intensive chemotherapy, regardless of the blast percentage. Clear, literature-based evidence for this clinical decision-making point would be highly valuable.
3.Incorporating the Role of CPX-351 in Intensive Chemotherapy
Considering that MECOM-rearranged AML represents a high-risk group often associated with secondary AML, we propose incorporating a dedicated section or subsection analyzing the role and efficacy of CPX-351 (Liposomal Cytarabine and Daunorubicin) in this patient population.
Please investigate and cite data, if available, from subgroup analyses of major CPX-351 clinical trials (e.g., in elderly tAML/sAML) that specifically address outcomes for $MECOM$ rearranged patients. This discussion is critical for guiding the optimal choice of induction therapy in high-risk patients.
4.Reinforcing the Scientific Basis for Novel Compound ReportsThe section on novel small-molecule compounds is exciting; however, the citations rely heavily on conference abstracts. To enhance the academic credibility of this review, we strongly recommend ensuring that the efficacy data for these novel compounds are primarily supported by peer-reviewed original research articles.
Please search for and utilize original papers detailing the mechanism of action, pre-clinical models, and early clinical data. Supplementing or replacing the abstract citations with high-impact primary literature will significantly strengthen the reliability and persuasiveness of this section.
None
Author Response
Comments 1: Strengthening the Discussion on Diagnostic Methods and the Risk of Misdiagnosis
Given the severe prognosis of MECOM-rearranged AML, accurate and timely diagnosis is paramount. We kindly suggest adding a more detailed discussion on the utility, limitations, and detection sensitivity of various diagnostic methods, including Conventional G-banding, FISH, RT-PCR, and Next-Generation Sequencing (NGS). Specifically, please discuss the risk of missing cryptic fusions and propose an integrated diagnostic strategy or algorithm that combines these methods to ensure comprehensive detection in clinical practice.
Response 1: We thank the Reviewer for this insightful comment. We have discussed the diagnostic modalities in brief and presented a stepwise approach for comprehensive evaluation.
“Accurate diagnosis of MECOM-rearranged AML relies on the detection of the lesion and thus a combination of cytogenetic and molecular methods. Conventional G-banding analysis usually represents the first step in detecting structural cytogenetic patterns like inv(3) or t(3;3). However, this method is limited to detection of aberrations at least 1 Mb in size and cryptic structural variants may be missed (37). In one study, pericentric inv(3) was missed in 16 of 17 cases by conventional karyotyping and was subsequently detected using metaphase Fluorescence in Situ Hybridization (FISH) (36). FISH testing, particularly with a MECOM break-apart probe (irrespective of the partner gene involved), is the most used adjunct to routine karyotype analysis (38). It enables detection of cryptic or atypical variants, complex chromosomal rearrangements that appear as unidentifiable marker chromosomes, and MECOM amplifications such as segmental duplications (6,39,40). However, FISH is limited to predefined loci and may not identify novel fusion partners. Polymerase chain reaction (PCR)–based assays, including allele-specific PCR, real-time PCR (qPCR), and digital PCR (dPCR), provide greater sensitivity, with dPCR capable of detecting variants at levels as low as 10⁻⁴ (41). Advances in next-generation sequencing (NGS) technologies now offer the most comprehensive approach, allowing identification of both canonical and novel MECOM fusion genes, as well as cryptic 3q26 rearrangements. With error-correction methods such as unique molecular identifiers (UMIs), NGS sensitivity can approach 10⁻⁶, minimizing false positives in low variant allele frequency (VAF) settings (41). NGS remains costly, time-consuming, and dependent on sophisticated bioinformatics pipelines, but improvements in these domains over time may yield more prevalent use.
In practice, a step-by-step diagnostic strategy is recommended, particularly in resource-limited settings where it may not be feasible to utilize all the above diagnostic modalities. This approach involves initial screening with G-banding analysis, followed by reflex MECOM FISH for suspected or inconclusive cases, and confirmation with qPCR or RNA-based NGS to define fusion partners and concurrent mutations for accurate risk stratification and therapeutic planning.”
Comments 2: Clarifying the Clinical Significance of the Blast Percentage Discrepancy (WHO/ICC)
The manuscript touches upon the AML classification criteria regarding the blast percentage threshold (10% in the selected text). We encourage you to provide a more in-depth discussion on the clinical implications and prognosis of $MECOM$-rearranged cases with blast counts below 10% (e.g., MDS/AML-MDS overlap). Based on existing literature, please articulate the rationale for whether these cases behave differently from typical AML (blast $\ge 10\%$) or if the underlying poor pathobiology warrants uniform treatment with intensive chemotherapy, regardless of the blast percentage. Clear, literature-based evidence for this clinical decision-making point would be highly valuable.
Response 2: We thank the Reviewer for this insightful comment. This is a very important point since there have been conflicting reports in literature regarding the prognostic significance of blast percentage in MECOM rearranged AML. A recent study showed patients with <20% bone marrow blasts had a longer median overall survival compared to patients with ≥20% blasts, whereas earlier registry studies suggested otherwise, showing no OS difference between <20% and ≥20% groups. We believe that although these thresholds do not have any implication on diagnostic purpose, they do behave differently from biological standpoint with higher blast count associated with more aggressive disease. We have provided evidence in our manuscript that, despite any differing biology, the therapeutic response to intensive therapies or hypomethylating agent-based regimens remain overall similar.
“The reduction or removal of blast percentage thresholds in recent AML classifications, including the ICC and WHO frameworks, reflects the recognition that MECOM-rearranged myeloid neoplasms exhibit aggressive biology and a high propensity for leukemic transformation, even at low blast counts (33–35). Although earlier studies demonstrated no significant survival difference between patients with <20% (previously classified as MDS) and ≥20% blasts—supporting the removal of the arbitrary blast cut-off—a recent study suggests that blast percentage retains prognostic significance within MECOM-rearranged AML (35–37). In this analysis, patients with ≥20% blasts had a composite complete remission (CRc) rate of 57% versus 45% for those with <20% blasts (p=0.31), and a median overall survival (OS) of 17 months compared with 9 months (p<0.01). Notably, cases with <5% blasts at presentation showed a median OS of 63 months. When stratified by <10%, 10–20%, or ≥20% blasts, outcomes were similar between the <10% and 10–20% groups (p=0.84), but patients with ≥20% blasts had significantly inferior OS compared with both <10% (p=0.0017) and 10–20% (p=0.0022) cohorts (37). These findings indicate that, while the blast percentage threshold may no longer serve a diagnostic purpose, higher blast counts within MECOM-rearranged AML are associated with more aggressive disease biology and poorer survival outcomes.”
“In a study involving 152 patients with MECOM-rearranged AML, baseline bone marrow blast percentage was found to be significantly associated with overall survival. The most favorable outcomes were observed in patients with non-inv(3)/t(3;3) 3q26 AML and blast counts below 20%. Overall survival was significantly better in the 3q26 AML group with <20% blasts compared to those with ≥20% blasts (p=0.0007) and to patients with inv(3)/t(3;3) AML and ≥20% blasts (p=0.0001), but was comparable to that of inv(3)/t(3;3) AML with <20% blasts (p=0.30). Intensive chemotherapy was associated with a higher hazard of death compared to low-intensity therapy, although the lower bound of the confidence interval approached unity. The addition of venetoclax did not significantly affect overall survival (37).”
Comments 3: Incorporating the Role of CPX-351 in Intensive Chemotherapy
Considering that MECOM-rearranged AML represents a high-risk group often associated with secondary AML, we propose incorporating a dedicated section or subsection analyzing the role and efficacy of CPX-351 (Liposomal Cytarabine and Daunorubicin) in this patient population. Please investigate and cite data, if available, from subgroup analyses of major CPX-351 clinical trials (e.g., in elderly tAML/sAML) that specifically address outcomes for $MECOM$ rearranged patients. This discussion is critical for guiding the optimal choice of induction therapy in high-risk patients.
Response 3: As of the October 2025, no prospective clinical trial or published real-world studies have specifically evaluated CPX‑351 (Vyxeos) in the subgroup of AML with MECOM rearrangements. Available evidence present indirect observations for broader ELN adverse risk and secondary AML—including therapy-related AML (t‑AML) and AML with myelodysplasia-related changes (AML‑MRC)—in which MECOM‑rearranged disease may have occurred but was not separately analyzed. In the pivotal phase III trial, AML with inv(3)(q21q26.2)/t(3;3)(q21;q26.2) or MECOM rearrangement was not separately included for evaluation. Several real‑world studies from Italy (n = 513), Germany (n = 188), and UK (n=353) confirmed the survival benefits of CPX‑351 in high‑risk AML, but none provide details separately for MECOM rearranged AML. In the French cohort (n = 103), only one of six patients harboring an EVI1 mutation achieved CR/CRi, and the mutation was significantly associated with a lower overall response rate (p = 0.03). Since we do not have any clinical data to address this important question, we have not included this in our manuscript.
References:
Lancet JE, et al. CPX-351 (cytarabine and daunorubicin) liposome for injection versus conventional cytarabine plus daunorubicin in older patients with newly diagnosed secondary acute myeloid leukemia. Journal of Clinical Oncology. 2018 Sep 10;36(26):2684-92.
Guolo F, et al. CPX-351 treatment in secondary acute myeloblastic leukemia is effective and improves the feasibility of allogeneic stem cell transplantation: results of the Italian compassionate use program. Blood Cancer Journal. 2020 Oct 6;10(10):96.
Rautenberg C, et al. Real-world experience of CPX-351 as first-line treatment for patients with acute myeloid leukemia. Blood Cancer Journal. 2021 Oct 4;11(10):164.
Legg A, et al. Real-world experience with CPX-351 treatment for acute myeloid leukemia in England: an analysis from the National Cancer Registration and Analysis Service. Clinical Lymphoma Myeloma and Leukemia. 2023 Oct 1;23(10):e323-30.
Chiche E, et al. Real-life experience with CPX-351 and impact on the outcome of high-risk AML patients: a multicentric French cohort. Blood Advances. 2021 Jan 12;5(1):176-84.
Comments 4: Reinforcing the Scientific Basis for Novel Compound Reports
The section on novel small-molecule compounds is exciting; however, the citations rely heavily on conference abstracts. To enhance the academic credibility of this review, we strongly recommend ensuring that the efficacy data for these novel compounds are primarily supported by peer-reviewed original research articles. Please search for and utilize original papers detailing the mechanism of action, pre-clinical models, and early clinical data. Supplementing or replacing the abstract citations with high-impact primary literature will significantly strengthen the reliability and persuasiveness of this section.
Response 4: We thank the Reviewer for this valuable suggestion. We have incorporated peer-reviewed original research articles wherever available to strengthen this section. Specifically, we supplemented the abstracts cited in references 60 and 63 with full peer-reviewed publications now listed as references 61 and 62, respectively. However, for certain emerging molecular targets discussed in Reference 64, data specific to MECOM-rearranged AML remain limited. Similarly, information on AB8939 is currently available only through conference abstracts and company press releases. The BRG1/BRM inhibitor data relevant to MECOM-rearranged AML have thus far been published only in abstract form (2025), although original article exists for studies in FLT3-mutated AML. Hence, although it would have been preferable to cite original articles, these findings nonetheless highlight the ongoing development and active research interest in the field of MECOM-rearranged AML.